# A Model to Evaluate the Flooding Opportunity and Sustainable Use of Former Open-Pits

## Izabela-Maria Apostu *, Maria Lazar and Florin Faur

Department of Environmental Engineering and Geology, Faculty of Mining, University of Petrosani,
332006 Petroșani, Romania; marialazar@upet.ro (M.L.); florinfaur@upet.ro (F.F.)
* Correspondence: izabelaapostu@upet.ro; Tel.: +40-728-740-003

**Abstract:** As a result of open-pit mining exploitations, impressive size gaps occur in the landscape. Their flooding leads to the occurrence of so-called open-pit lakes and represents an interesting way to reclaim and use sustainably the degraded land. In the literature, there are numerous plans, strategies, and guidelines for mine closure and open-pit recovery, but these are usually developed at the regional or national level and offer general suggestions, which must be evaluated and approached case-by-case. Because there is still no way to evaluate the opportunity of flooding the open-pits, a methodology for assessing this opportunity was developed to identify the open-pits that are suitable for flooding, this being the main objective of the paper. The paper is novel because of the multicriteria evaluation of open-pits and their remaining gaps, the logical succession of the criteria, and the proposed concept, methods, models, and equations that allow a complex assessment of the flooding opportunity. The methodology also aims to ensure maximum safety conditions in the former mining perimeter, the socio-economic and cultural requirements of local communities, the harmonization of the land in accordance with adjacent ecosystems, and the sustainable development of the region.

**Keywords:** open-pits; flooding; opportunity assessment; model; methodology; evaluation matrix; sustainable development

## 1. Introduction

Sustainable development is a relatively new concept, put forward for the first time in the Brundtland Report (also known as "Our common future"), in 1987. Sustainable development is defined as "development that meets the needs of the present generations without compromising the ability of future generations to meet their own needs." [1]. Environmental degradation was one of the main issues recognized at the global level. Open-pit mining activities are among the ones with a major impact on the environment. The direct impact is manifested by the degradation of the land, the local ecosystem, and by the pollution of the environmental components. Some of the negative effects are also felt in the post-closure period, in the case of abandoned mining perimeters, and they persist until nature manages to recover on its own.

Open-pit lakes are simple bodies of water, formed through natural flooding (inflow of water from precipitation or aquifers) and/or artificial flooding (water adductions from water sources located in the adjacent areas) of the gaps resulting after the cessation of the open-pit exploitation activity.

Former open-pits can be recovered and reused for various purposes, such as landfills, off-road circuits, amphitheaters, pit lakes that can take over various functions, but based on a well-defined methodology and representative criteria can be chosen the most suitable reuse option [2].

When there is a possibility to flood the remaining gaps after the cessation of the mining activity, giving rise to the so-called open-pit lakes, several possibilities of reuse occur. The most common uses of the open-pit lakes are lakes for recreation, lakes for fish farming, natural lakes, potable or industrial

water reservoir, irrigation water tank, retention basin for protection against floods, etc. Regardless of the type of reuse, the open-pit lakes offer a number of benefits from an ecological and economic point of view.

The process of planning, recovery, and reuse of open-pit degraded lands is important as it ensures the sustainable development of a former mining region. It accelerates the natural restoration process. It must take into account environmental aspects and the socio-cultural requirements of the population. The correct rehabilitation of the former open-pits and, in general, of former mining areas, must tend towards a clear objective: naturalistic area, recreational area, industrial area, etc. Multiple land reuse is a relatively new practice that involves combining different forms of use, which complement each other [2,3].

Worldwide, thousands of lakes were created in former open-pit mines after the cessation of the mining activity. Water quality varies among pit lakes due to geological and hydrogeological conditions. In many cases the lake water is toxic, containing a high concentration of heavy metals, acidic or saline, posing risks to adjacent communities and ecosystems, while in other cases the lake contains fresh and clean water [4–9]. Besides the water quality, which is an important aspect when planning the reuse of the pit lake, the geotechnical risk arises. The geotechnical risks refer to hazards such as landslides, which may have major impacts on the environment and local communities [5,8–12].

Numerous plans and strategies have been developed to establish the most efficient methods to recover and reuse the open-pit mining degraded lands and the remaining gaps resulting at the end of the exploitation activity. McCullough is one of the researchers who is constantly focusing on the issue of reuse of former open-pits and pit lakes, on water quality of pit lakes, and their impact on the environment and local communities. Through his numerous collaborations [13–21], he analyzed the key issues (especially risks), opportunities for sustainable mining, beneficial end-uses of pit lakes, and suggested best available practices in mine closure planning for pit lakes. Guidelines for mine closure and reclamation are usually elaborated at a regional or national level, offering general advice and solutions which must go professional interpretation and a case-by-case approach [13,20,22–24]. Schultze et al. [25] reported the German experiences regarding the flooding of pits resulting from the exploitation of lignite from the last 20 years. They analyzed the important contribution of groundwater to the flooding of former open-pits and the benefits of the natural flooding process, highlighting the fast flooding and the advantages regarding the stabilization of the pit walls.

Beneficial end-uses of former open-pits refer to that type of use that provides economic, health, welfare, or safety benefits to the community and ecological benefits to the environment. Johnson and Wright [26], used an existing risk management guideline and developed a semi-quantitative method to assess the risk on the environment from different beneficial end-uses of pit lakes. So, risks need to be balanced against economic and social benefits.

The main objectives of the paper are establishing the evaluation criteria to ensure a complex assessment of the flooding opportunity of the remaining gaps for the sustainable reuse of the mining degraded lands taking into account the most important aspects and elements that may influence this process, development of the flooding opportunity evaluation matrix and verifying the usefulness of the proposed methodology on the specific case of the Rovinari Mining Basin open-pits and their remaining gaps. The methodology was designed to ensure maximum safety conditions in the former mining perimeter, the harmonization of the land in accordance with adjacent ecosystems, the sustainable development of the region taking into account the socio-economic and cultural requirements of local communities.

## 2. Materials and Methods

By starting from a good knowledge of the flooding process, the criteria that influence this process have been identified. These criteria were described and analyzed in detail, because based on them, the matrix for assessing the flooding opportunity of the remaining gaps, was developed. Thus, an original methodology is presented for establishing the extent to which the flooding of the remaining gaps of lignite open-pits is an opportune measure for the rehabilitation of degraded lands.

### 2.1. Assessing Criteria

Taking into account the characteristics and aspects that allow the separation and even elimination of the pits that are not suitable for flooding, the following assessing criteria have been established for the opportunity of flooding former open-pits:

- Geomorphology and orography of the area;
- Configuration of the remaining gap;
- Necessity of appearance of a water body in the area;
- Necessity to restore the aquifer resources;
- Hydrology and hydrogeology of the region;
- Stability conditions of the final slopes of the remaining gap;
- Accessibility and distance to the areas of interest;
- Investment for the recovery and rehabilitation of the remaining gap;
- Population requirements.

For abandoned open-pits that have been partially reintegrated into the landscape naturally, it is only possible to carry out maintenance and support works for the acceleration and harmonious reintegration of the degraded land into the landscape.

### 2.1.1. Criterion C1—Geomorphology and Orography of the Area

The studies regarding the geomorphological and orographic conditions provide information on the characteristics of the site, such as relief forms, altitude, inclination, their way of grouping, and spreading in the territory in which the useful mineral substances deposit was formed. Depending on the site-specific relief forms, the exploitable deposit may be concentrated: in hilly areas (Figure 1a,b), in hilly and meadow areas (Figure 1c), or in meadow areas (Figure 1d).

So, it is possible to assess the possibility of occurrence of a remaining gap in the mining perimeter taking into account, in particular, the development of the useful mineral substances deposit and the characteristic forms of relief.

### 2.1.2. Criterion C2—Configuration of the Remaining Gap

The excavation, transportation, and dumping technologies applied in the open-pit mining perimeters determine the configuration of the remaining gap, respectively the final shape and geometric elements of the final slopes. Depending on the site's relief forms, the position of the deposit relative to the surrounding land, the volume of the inner dump related to the volume of the remaining gap there are four situations (Figure 2a,d). The depth can be measured relative to the level of the surrounding land or the upper platform of the inner dump if this exceeds the level of the surrounding land (Figure 3).

It is important to know the configuration of the remaining gap, to determine the floodable volume of the open-pit and the required amount of water.

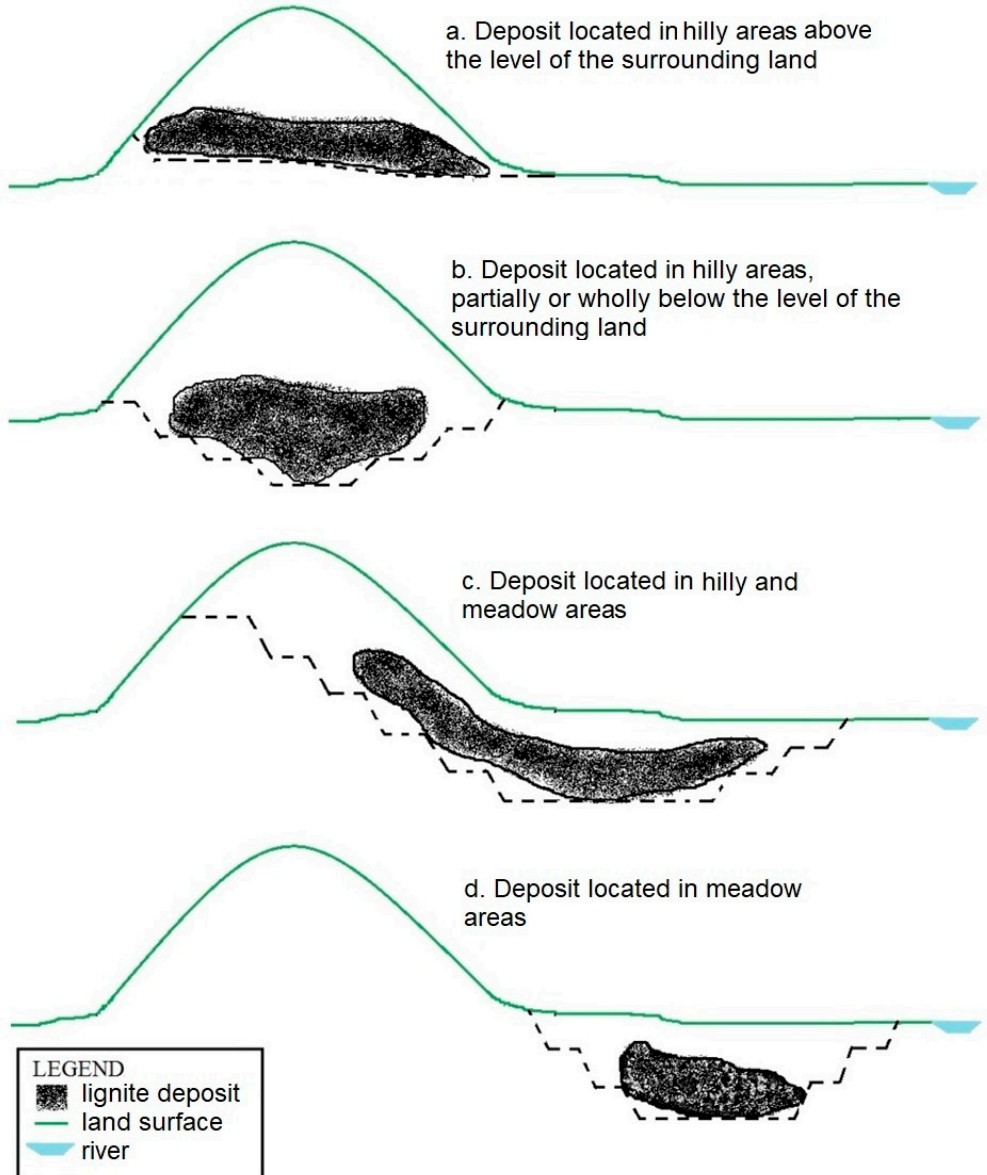

**Figure 1.** The shape of the open-pit according to the location of the deposit in relation to the forms of relief [27].

### 2.1.3. Criterion C3—Necessity to Restore the Aquifer Resources

In many cases, the open-pit exploitation of useful mineral substances can be achieved only under the conditions of natural drainage (through quarry slopes) or artificial drainage (through dewatering drillings) of aquifer formations in the mining perimeter.

The effect of dewatering works is manifested by a quantitative impact on phreatic or deep aquifers, these works leading to the expansion of depression curves and the decrease or even disappearance of groundwater resources, with direct negative effects: the drying of the wells, reduction of water intake flow, damage to vegetation, etc. [28].

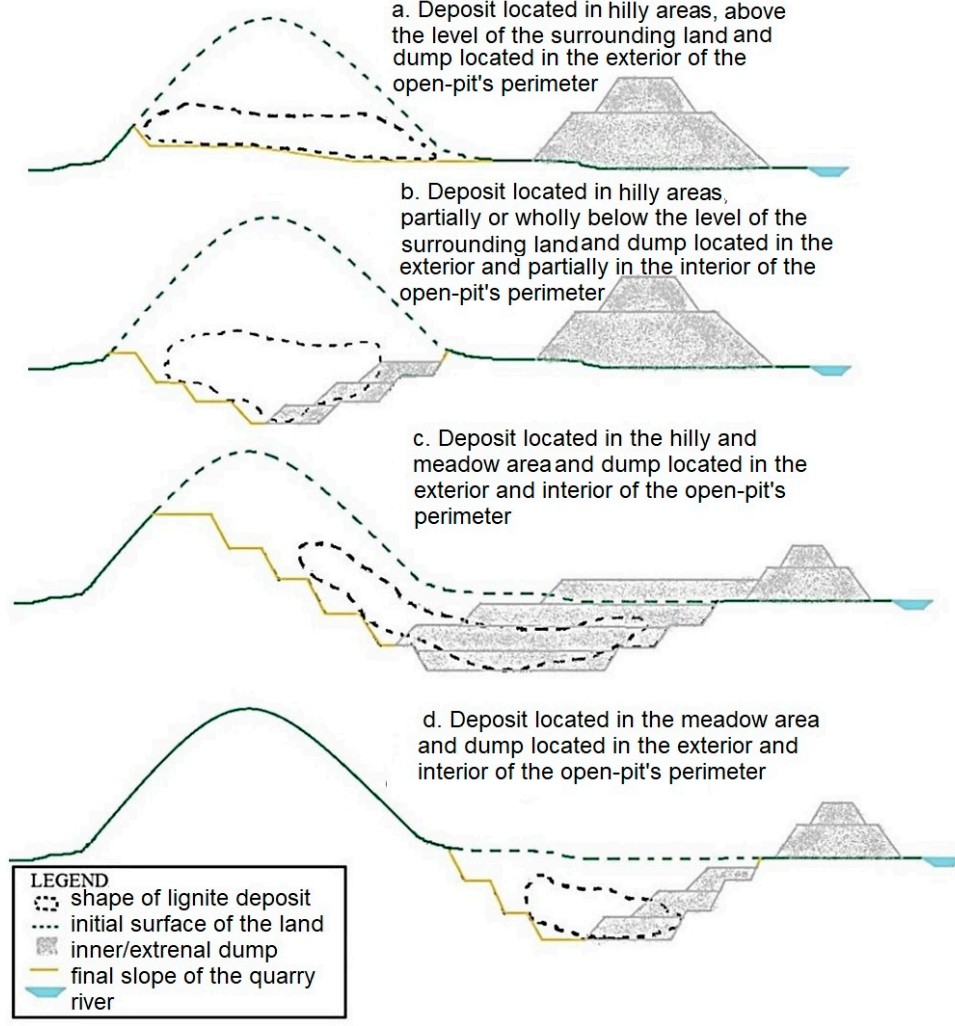

**Figure 2.** The shape of the remaining gaps according to the location of the deposit and the way of construction of the inner and external dumps [27].

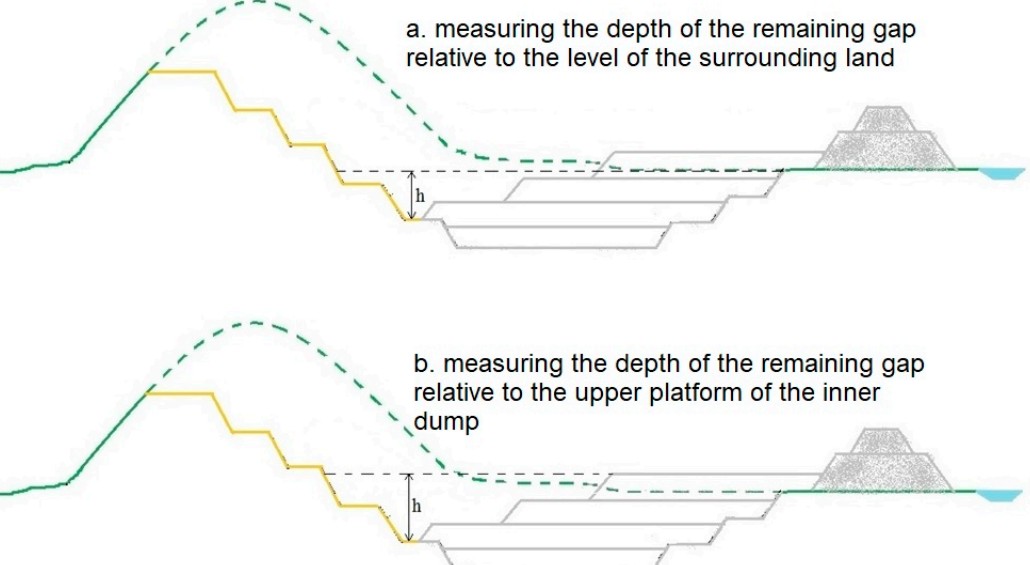

**Figure 3.** Measuring the depth of a remaining gap [27].

If the natural restoration of aquifers isn't possible, the water requirements of each type of land use shall be taken into account for the assessment of the necessity to restore groundwater resources which largely depends on the type of use of adjacent land/lands.

To prevent and reduce the negative effects of drought, programs should be drawn up which include measures to give priority to the supply of potable water and the irrigation of crops [29].

Depending on the type of use and water requirements, the lands were classified into 9 categories, ranked and evaluated by grading them with grades (*x*) from 1 to 9. It is also important to know the priorities in terms of potable water supply. Thus, a hierarchy of priorities was established according to the type of land use and the values of a priority coefficient (*c*) were defined from 1 to 4 (Figure 4).

| | |
|---|---|
| Agricultural areas (*x* = 9) | • cultivated land, pastures, orchards, vineyards etc., including small households and farms |
| Urban areas (*x* = 8) | • cities, villages, institutional centers, shopping centers, parks and recreation areas, cemeteries, landfills etc. |
| Industrial areas (*x* = 7) | • mining perimeters, deposits, other related constructions; industrial parks; industrial complexes etc. |
| Protected areas (*x* = 6) | • parks and natural reservations |
| Natural areas (*x* = 5) | • grassy meadows, meadows with shrubs, natural areas of plain, hill or mountain |
| Forested areas (*x* = 4) | • deciduous, coniferous or mixed forests |
| Lakes and river areas (*x* = 3) | • streams, rivers, lakes etc. |
| Poor lands (*x* = 2) | • lands with a very thin layer of vegetal soil, sandy, arid, rocky etc. |
| Transport, communications and utilities (*x* = 1) | • streets, highways, railways, including airports, stations, parking lots; water, gas, electricity, networks etc. |

**Figure 4.** Types of land use.

Thus, grade 1 is given to land occupied by transport, communications, and utilities as it does not involve water consumption (in their exploitation period), and grade 9 is given to agricultural land that has the highest requirements for water. The highest value for the priority coefficient belongs to the land/type of use which, in case of drought, depends on the potable water supply, so it has priority.

For the evaluation of adjacent lands with multiple uses, the weighted average (Equation (1)) will be calculated, taking into account the value of the land and the priority coefficients depending on the water requirements of the land, respectively the potable water supply priority (Table 1).

$$M_p = \frac{\sum x_i \cdot c_i}{\sum c_i} \tag{1}$$

where:

*x*—the value of the land (grade); *c*—the priority coefficient; $M_p$—the weighted average.

**Table 1.** Potable water supply priority.

| Type of Land Use | Priority Coefficient, $c$ |
|---|---|
| Urban areas | 4 |
| Agricultural areas | 3 |
| Industrial areas | 2 |
| Protected areas, natural areas, forested areas, lake and river areas, poor lands, transport, communications, and utilities | 1 |

The weighted average ensures a proper assessment of the necessity to restore aquifer resources. Thus, given that land has multiple uses, the weighted average allows obtaining a value favorable to the type of use with higher requirements for water in general, respectively for potable water.

### 2.1.4. Criterion C4—Necessity of Appearance of a Water Body in the Area

The flooding method of former open-pits involves the appearance in the area of a water body, respectively a new aquatic ecosystem whose structure depends on the type of future use, in a territory that previously did not meet such conditions.

Taking into account the specifics of the area and the development strategy of the region, the occurrence of a water body can have a major significance. For example, in an agricultural area, such a lake can be used as a water reservoir for irrigating crops during dry periods. If the regional development strategy aims at tourism development, the lake can be arranged to carry out recreational and leisure activities, a type of use that can also ensure the economic development of the region. On the other hand, the lake can be populated with fish to support sustainable fishing (fish farming) or to practice recreational sport fishing. If the area of activity specific to a region is hunting, the creation of a water body is not strictly necessary, but the occurrence of a lake in the landscape does not affect the way this activity is carried out [7].

The necessity for the occurrence of a water body in a given region can be assessed in terms of the characteristic areas of activity in the immediate vicinity of the mining perimeter and the water requirements specific to each area (Figure 5).

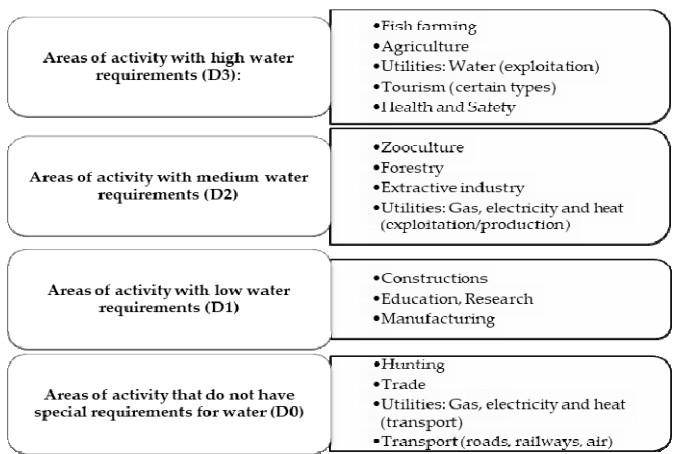

**Figure 5.** Water requirements according to specific areas of activity.

### 2.1.5. Criterion C5—Hydrology and Hydrogeology of the Region

The presence of permanent water sources, such as rainwater, aquifers, and, less frequently, surface watercourses (streams, rivers, etc.) are considered of major importance when applying the flooding method, as it ensures the flooding of the open-pit naturally and in a shorter period of time. Natural flooding involves lower costs, with no additional water supply being required.

The supply of an open-pit lake is generally achieved by water inflow from underground infiltrations and precipitations, and the discharge of water from the lake is achieved by natural drainage and evaporation process. In this way, a balance is ensured between the supply and the water losses of the newly formed lake.

Depending on the geographical location, respectively the specific climatic conditions are given by the amount of precipitation water (precipitation water is noted $Q_P$; $Q_P \geq 0$) that reaches the lake and the amount of water evaporated from the surface of the lake (evaporated water is noted $Q_E$; $Q_E > 0$), the amount of water that actually contributes to the flooding of the open-pit can be determined. Depending on the ratio between the amount of precipitation and the amount of evaporated water, there are several situations [27]:

- $Q_P/Q_E > 1$—precipitations has an insignificant ($Q_P/Q_E = 1 \div 1.1$), or a significant contribution (if $Q_P/Q_E > 1$) to open-pit flooding;
- $Q_P/Q_E = 1$—there is no discharge or charge of the lake;
- $Q_P/Q_E < 1$—relatively low discharge of water from the lake. Another situation can occur, respectively when $Q_P \to 0$ and $Q_E > 0$ results a significant discharge of the lake.

The presence of aquifer resources is essential for the formation of an open-pit lake, naturally, as in most cases, the inflow of water from rainfall is insufficient. If the influx of groundwater can significantly contribute to the flooding of the gap, then the creation of an open-pit lake is an opportune choice. Otherwise, it is recommended to choose another type of reuse.

Over time, numerous studies have been conducted to assess the degree of difficulty of exploitation of the deposits depending on the hydrogeological conditions specific to the region. According to a complex classification existing in the literature, exploitable deposits can be classified hydrogeologically into four classes [30].

In the exploitation stage, large water inflows are not desirable, as they make exploitation and dewatering works difficult, involve high financial costs and risks regarding the safety of employees and equipment; however, in the stage of recovery and reintegration into the landscape, by creating a lake, these hydrogeological characteristics are favorable, as they ensure the restoration of aquifer resources and the flooding of the gap naturally in a relatively short period of time, without additional costs. In this sense, an analogy was made between the degree of difficulty of the exploitation of the deposits depending on the characteristic hydrogeological conditions and the possibilities of flooding the former open-pits. Therefore, the more difficult the hydrogeological conditions during exploitation and the higher the groundwater inflows, the higher the possibilities of flooding the open-pit after the cessation of the exploitation activity.

In addition, it is possible to evaluate the storage and release capacity of water by rocks, the possibility of flooding, and maintaining a constant level of water in the lake, the speed of water discharge from the lake through an important physical characteristic, namely permeability of rocks.

Non-cohesive rocks with large granulation drain more easily (sand, gravel), while pseudo-cohesive rocks with fine granulation (dust, clay) release water very hard [30].

### 2.1.6. Criterion C6—Stability Conditions of the Final Slopes

In general, the former open-pits are bordered by the final in-situ and inner dump slopes. Decommissioning the dewatering systems, restoring the aquifer resources, and raising the water level in the lake, influences the stability of the slopes negatively, by manifesting the pore water pressure and/or the hydrodynamic pressure under the conditions of formation of aquifers currents, but also positively by the water pressure manifested on the surface of the slope. Knowing the behavior of rocks in contact with water and the geotechnical risks that may occur in the newly created conditions is essential for increasing the security and safety of objectives and people in areas of influence [31,32].

It is important to assess the stability of the final slopes before the flooding and to identify all the influencing factors (the influence of water, overloads, seismic shocks, etc.) to be able to estimate the stability reserve trend, over time.

Through the excavation, transport, and dumping processes, the strength characteristics of the rocks suffer important changes, so the dump slopes are characterized by lower stability reserves compared to in-situ slopes. When assessing the opportunity of flooding an open-pit according to the stability conditions of the final slopes, the unfavorable situations are taken into account.

The optimal value of the safety factor for earthworks with a long residence time is recommended to be between 1.25 ÷ 1.5. Depending on the degree of stability, waste rock deposits (dumps) were classified into four categories [33]. The imposed value of the safety factor needs to be higher than 1 as the time factor cannot be counted. The negative influences will determine in time the worsening of the strength characteristics and implicitly the reduction of the stability reserve.

### 2.1.7. Criterion C7—Accessibility and Distance to the Areas of Interest

In the stage of recovery and reconstruction of degraded lands, the existence of permanent roads and short access roads to the objectives of interest (such as utilities, cities, etc.) is an advantage, primarily in financial terms. If the type of reuse requires the frequent presence of staff and/or visitors, it is necessary to connect to existing transport routes and build appropriate infrastructure.

In most countries, road networks reflect the development of a hierarchy of roads, with highways at the highest level and local access roads at the lowest. The structure of the road (unpaved, paved, with semi-permanent pavements and with permanent pavements) is established depending on the distance between the objectives, the traffic regime (open or closed to public traffic), and traffic intensity (very intense to reduced traffic intensity) [34].

If the identified roads are of the same rank and have the same condition, the differentiation of the score can be done according to other aspects, such as the length of the roads that make the connection between the open-pit and the main road.

It is important to assess the possibility of reusing an open-pit taking into account the distance from the areas of interest (cities, agricultural areas, etc.), as the advantages are greater the shorter is this distance.

Naturalistic recovery is a type of reuse that is suitable for degraded lands located at great distances from urban centers and main roads, while recreational, residential recoveries are suitable for degraded lands located at relatively short distances from areas of interest and road [35].

### 2.1.8. Criterion C8—Investments for Land Recovery and Rehabilitation

The existence of natural sources of flooding, the short distance from the areas of interest, the existence of permanent access roads, and constructions that can be used for other purposes have important financial advantages.

Based on a classification existing in the literature [36], a new classification was developed adapted to the aim of the paper to ensure the evaluation (subjective, but which provides sufficient information) of the necessary investments regarding the recovery, rehabilitation, and restoration of mining degraded lands and former open-pits:

- Insignificant investments—flooding, recovery, and reintegration into the landscape occur naturally within an acceptable period of time.
- Reduced investments—flooding, recovery, and reintegration into the landscape occur naturally, but some anthropogenic interventions are needed to accelerate the processes: leveling and resloping, use of existing objectives, etc.
- Medium investments—flooding, recovery, and reintegration into the landscape are done naturally and anthropically through the water supply, being necessary other anthropic

interventions: resloping, leveling, compaction, remodeling, accelerating the process of revegetation, reconstruction, transformation or development of existing objectives, etc.

- High and very high investments—flooding, recovery, and reintegration into the landscape are mostly anthropogenic: water supply, resloping, leveling, compaction, remodeling of banks, revegetation, construction of new objectives, etc.

### 2.1.9. Criterion C9—Population Requirements

In the process of planning the use of degraded land, it is necessary to involve all stakeholders: state, local authorities, local communities, etc., to identify and establish uses and infrastructures appropriate to the type of land reuse. The demands of the population reflect the individual needs and priorities of local communities [35].

The evaluation of population requirements can be done by different methods, one of the effective methods being the evaluation by opinion poll. Opinion polls should be aimed strictly at subjects who can benefit from the recovery and reuse of the mining degraded land, as well as researchers in the field.

Based on the results of the opinion polls and the hierarchy of population requirements, the method described below is applied to determine the score awarded according to this criterion, respectively to assess the flooding opportunity [37]:

- the value of a constant (*c*) is determined using Equation (2);
- the final score is determined using Equation (3);

$$c = \frac{P_{max}}{n_{var} - 1}, \tag{2}$$

$$P_f = (n_{var} - p_{lake}) \cdot c \tag{3}$$

where: $P_{max}$—maximum value of the score (in the proposed methodology the maximum value is P = 3); $n_{var}$—the number of reuse variants for the open-pit (offered in the questionnaire); $p_{lake}$—the position of the "open-pit lake" variant in the hierarchy of population requirements; *c*—constant; $P_f$—final score.

### 2.2. Evaluation Matrix

The 9 criteria established and described are the basis for the elaboration of a complex evaluation matrix that ensures the obtaining of results with a high degree of confidence, which makes the process of recovery and rehabilitation of a remaining gap to follow the optimal direction both ecologically and economically.

Depending on the described criteria, to evaluate the opportunity of flooding former open-pits, appropriate scores (P) were established with values between 0 and 3, where P = 0 characterizes the inopportunity of flooding of an open-pit, while P = 3 characterizes the major opportunity of flooding of an open-pit.

Table 2 shows the matrix for evaluating the opportunity of flooding of former open-pits, which are centralized the defined assessing criteria, respectively the scores according to different characteristics. To develop the matrix, numerous existing classifications in the specialized literature were taken into account, but also a series of evaluations and personal assessments made based on studies and researches conducted in the fields of Engineering, Environmental Protection and Mines, Oil, and Gases.

**Table 2.** The matrix of evaluation of the opportunity of flooding of former open-pits.

| Score Criterion [1] | P = 0—Inopportune | P = 1—Reduced Opportunity | P = 2—Average Opportunity | P = 3—High Opportunity |
|---|---|---|---|---|
| C1 | hilly or mountain area, the deposit is above the level of the surrounding land, practically does not result in a gap | hilly (or hilly and meadow area with extension to the hilly area) or mountain area, the deposit is partially or completely below the level of the surrounding land, reduced probability of occurrence of a remaining gap (it is likely to result in a remaining gap, but its dimensions are usually small) | hilly or hilly and meadow area, relatively high probability of occurrence of a remaining gap | meadow area, high probability of occurrence of a remaining gap |
| C2 | practically does not result in a gap, h = 0 m | shallow depth of the remaining gap, h = 0–10 m | medium depth of the remaining gap, h = 10–30 m | high depth of the remaining gap, h > 30 m |
| C3 | $M_p \le 2$; predominates lands without special water requirements and for which water supply is not a priority; it is not necessary to restore the aquifer resources; | $2 < M_p \le 5$; predominates lands with low water requirements, for which water supply is not a priority; the average need for restoration of aquifer resources; | $5 < M_p \le 7.5$; predominates lands with average water requirements, for which water supply is a priority; high need for restoration of aquifer resources | $7.5 < M_p \le 9$; predominates lands with high water requirements, for which water supply is a priority; major need for restoration of aquifer resources. |
| C4 | for the domain of activity that does not have water requirements (D0), there is no need for creating a lake | for the domain of activity that has low water requirements (D1), reduced need for creating a lake | for the domain of activity that has average water requirements (D2), the average necessity of creating a lake | for the domain of activity that has high demands on water (D3), a major need of creating a lake |
| C5 | $Q_P \to 0$, $Q_E = +$, it results in a significant discharge of the lake 1st class, mining perimeter with reduced possibility of flooding from aquifer formations; a mixture of aquifer rocks (sands, gravels, etc.) | $Q_P/Q_E < 1$, results in a relatively small discharge of the lake, but that can be covered by the influx of water from the aquifer formations class II, mining perimeter with the average possibility of flooding from aquifer formations; a mixture of predominantly aquiferous rocks | $Q_P/Q_E = 1$, results that rainfall does not contribute to the flooding of the remaining gap, but there is no loss of water from the lake class III, mining perimeter with a high possibility of flooding from aquifer formations; a mixture of predominantly aquiclude rocks | $Q_P/Q_E > 1$, results that rainfall has an insignificant ($Q_P/Q_E = 1 \div 1.1$), or a significant (if $Q_P/Q_E > 1$) contribution to flooding the remaining gap class IV, mining perimeter with a major possibility of flooding from aquifer formations; a mixture of aquiclude rocks (marls, clays, etc.) |
| C6 | 4th class of stability, Fs <1, unstable slopes, with active displacements; | 3rd class of stability, slopes with reduced stability / at the limit of stability, Fs ≈ 1, slopes that can enter dangerous movement even as a result of the individual action of some triggering factors (such as the presence of water in the body of the slope as a result of heavy rainfall, explosions, earthquakes, vibrations from the vehicles of high tonnage machines or overloads given by overloading the berms/platforms, etc.); | 2nd class of stability, slopes with high stability, Fs = 1.25 ÷ 1.5, slopes at which possible displacements can be recorded in case of concomitant or individual action of some triggering factors, but which can be limited by arrangements; | 1st class of stability, Fs > 1.5, slopes with high stability reserve, at which the probability of sliding is very low or even zero (only in case of the simultaneous action of several triggering factors can slip phenomena). |
| C7 | dirt roads improved or unimproved, temporary, strictly for driving vehicles, closed to public traffic, hardly accessible very long distance from the areas of interest (>50 km) | paved roads, opened or closed to public traffic, accessible, very low traffic relatively large distance from the areas of interest (10–50 km) | semi-permanent roads, opened to public circulation, easily accessible, low traffic or medium medium distance from the areas of interest (1–10 km); | permanent roads, opened to public traffic, easy access, heavy or very intense traffic low distance from the areas of interest (0–1 km) |
| C8 | high or very high investments | average investments | reduced investments | does not require investment or involves very small, insignificant investments |
| C9 | | $P_f = (n_{var} - p_{lake}) \cdot c$, (see Equation (3)) | | |

[1] C1–C9—the evaluation criteria.

The ecological benefits are not enough for the sustainable development of a region, so that appropriate research allows the choice of the most appropriate types of reuse, to complete the list of benefits from an economic, social, cultural, etc. point of view.

## 3. Results

Next, to verify the new methodology, elaborated for the evaluation of the flooding opportunity of the remaining gaps, the case of the lignite open pits from the Rovinari-Romania basin (the central coordinates of the Rovinari basin: latitude 44°88′41″ N–longitude 23°20′01″ E) was considered (Figure 6).

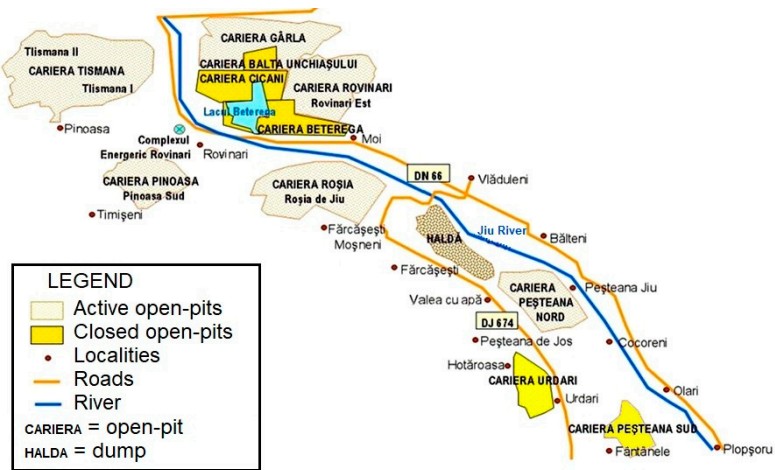

**Figure 6.** Rovinari Mining Basin [38].

Currently, 5 lignite open pits are active in this basin: Rovinari, Tismana, Pinoasa, Roșia de Jiu, and North Peșteana.

### 3.1. Assessing the Opportunity of Flooding the Lignite Open-Pits from Rovinari Mining Basin

Taking into account the established criteria and the complexity of the methodology and evaluation methods, the Rovinari Mining Basin open-pits were evaluated according to each criterion. It is worth mentioning that due to space reasons, the analyzes performed for each open-pit in relation to each criterion are not presented in detail.

### 3.1.1. Geomorphology and Orography of the Area (C1)

From a geomorphological point of view, in the analyzed region, units of the Subcarpathians and the Getic Plateau are known. The mixed relief (hilly and meadow) predominates in three perimeters: Rovinari, Tismana, and Roșia de Jiu, a meadow in the North Peşteana perimeter and hilly in the Pinoasa perimeter. The lignite deposits were completely or partially below the level of the surrounding land. The probability of resulting in a remaining gap after the cessation of mining activity is relatively high in 3 of the 5 mining perimeters analyzed.

### 3.1.2. Configuration of the Remaining Gap (C2)

The open-pits were evaluated according to their depth. For Pinoasa (current depth 80 m) and Roșia de Jiu (current depth 120 m) open-pits, the probability of occurrence of a remaining gap is relatively low due to the extension of the quarries in hilly areas, respectively the extension of the interior dumps, which by the end of the activity can considerably reduce the final depth of the gaps (given that the inner dump covers the base and part of the in-situ slopes of the quarry) or can completely fill the remaining gap. Consequently, for the 2 remaining gaps, it was estimated that the final depth will be

reduced, thus, from this point of view, presenting a reduced opportunity for flooding. For Rovinari (maximum depth 75 m), Tismana (maximum depth 50 m), and North Peşteana (maximum depth 80 m) open-pits the depths will not show significant variations until the cessation of the activity, so these values can be taken into account for evaluation [38].

### 3.1.3. Necessity to Restore the Aquifer Resources (C3)

To establish the necessity to restore the aquifer resources in the area of the Rovinari Mining Basin, the types of adjacent land use to each open-pit were studied and based on their value (x), water requirements, and water supply priority (defined through priority coefficient, c), the opportunity of flooding was assessed (Table 3).

**Table 3.** Necessity to restore the aquifer resources.

| Perimeter | Adjacent Areas | Value, x | Priority Coefficient, c | Weighted Average, Mp |
|---|---|---|---|---|
| Tismana | Industrial | 7 | 2 | |
| | Forested | 4 | 1 | |
| | Agricultural | 9 | 3 | 6.85 |
| | River | 3 | 1 | |
| Rovinari | Industrial | 7 | 2 | |
| | Forested | 4 | 1 | 7.5 |
| | Agricultural | 9 | 3 | |
| Pinoasa | Forested | 4 | 1 | |
| | Agricultural | 9 | 3 | 7.75 |
| | Urban | 8 | 4 | |
| Roșia de Jiu | Forested | 4 | 1 | |
| | Agricultural | 9 | 3 | 7.33 |
| | River | 3 | 1 | |
| North Peșteana | Agricultural | 9 | 3 | 7.5 |

### 3.1.4. Necessity of Appearance of a Water Body in the Area (C4)

The specific economy of Gorj County is industrial-agricultural. The economy of the city of Rovinari and the adjacent villages has promoted the extractive industries and electricity production by burning coal (lignite), to which agriculture is added. Agriculture is a non-performing economic branch at present because the agricultural area is of a medium quality and subsistence agriculture is practiced, with outdated technologies and equipment. Tourism is in its infancy but has appreciable development potential [39]. The mining activity in the region is coming to an end so agriculture and tourism are considered as the main economies of the future.

Given the major water requirements of crops and dry periods specific to the region, the occurrence of a water body has many benefits, such as being an additional source of water for crop irrigation [40].

### 3.1.5. Hydrology and Hydrogeology of the Region (C5)

To estimate the contribution of precipitation to the flooding of the open-pits, the values of average annual precipitation and potential evapotranspiration recorded in the Rovinari area were taken into account (Figure 7). The value of evapotranspiration is given by the sum of the amounts of water that reach the atmosphere in the form of vapors through the processes of evaporation and plant transpiration. So, using theUsing evapotranspiration in the calculations provides relative but satisfactory information.

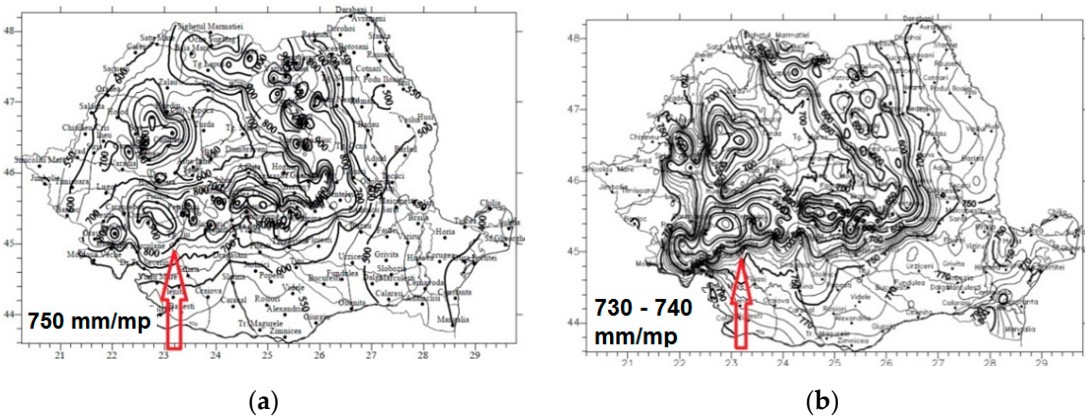

**Figure 7.** (**a**) Average annual rainfall; (**b**) average potential evapotranspiration [41].

According to the annual amounts of precipitation (750 mm/m$^2$) and potential evapotranspiration (730–740 mm/m$^2$) in the area, this results in a surplus of water of about 10–20 mm/m$^2$/year, valid for all five open-pits, given their location (relatively small distance between them and similar climatic conditions).

The development of lignite deposits in the Jiu River meadow area, respectively in a region with impressive aquifer horizons, has a positive influence on the restoration of aquifer resources and flooding of open-pits, as it ensures the development of these processes naturally, without involving major financial investments.

In the studied perimeters hydrogeological conditions are generally difficult and very difficult. The assessment of flooding opportunity was made taking into account the value of the water inflow coefficient, which varies from one perimeter to another as it follows: 2.6 m$^3$/t for Tismana, 3.7 m$^3$/t for Pinoasa, 5.41 m$^3$/t for Rovinari, 12,87 m$^3$/t for North Pesteana, and 16.32 m$^3$/t for Roșia de Jiu.

The geological formations consist of marls and clays (50–70%), sands (5–20%), and vegetal soil (5–20%), formations in which the lignite layers are incorporated [42]. Aquiclude rocks predominate.

### 3.1.6. Stability Conditions of the Final Slopes (C6)

Based on the research and documentation, observation, and information obtained in the field regarding the stability reserve of in-situ and dump slopes in the studied mining perimeters [38,43,44], it was found that, in general, the slopes are stable. Thus, it was found that all in-situ slopes and the slopes of the inner dumps of the Roșia de Jiu, North Peşteana, and Pinoasa open-pits present high stability reserves, while the slopes of the inner dumps of the Tismana and Rovinari open-pits are at the equilibrium limit. For the final assessment, the most unfavorable values were taken into account, which increases the degree of safety and security of the objectives in the area of influence.

### 3.1.7. Accessibility and Distance to the Areas of Interest (C7)

According to the road quality map in Romania, the access roads in the studied area are DN66 national road, with permanent asphalt, excellent condition, open to public traffic, easy access, intense traffic; DJ674 county road, with semi-permanent asphalt, mediocre condition, open to public traffic, easily accessible, low or medium traffic; DC73 communal road, with semi-permanent pavements, open to public traffic, low traffic. The connection with the open-pits is made through unpaved roads, difficult to circulate, closed to public traffic. The differentiation of the score was made taking into account the length of the connecting roads (DN66-Tismana perimeter ≈3 km; DN66-Rovinari perimeter ≈4–5 km; DN66-Pinoasa perimeter ≈4–5 km; DN66-Roșia de Jiu perimeter ≈1 km; DJ674-North Peşteana perimeter ≈0.5–1 km) and the shortest ones received a favorable score, as the costs of modernizing them are lower.

The smallest distances from the areas of interest for which the new land use can bring ecological and economic benefits were taken into account (Table 4).

**Table 4.** Distance to the areas of interest.

| Perimeter | Areas of Interest | Distance (km) |
|---|---|---|
| Tismana | | 5–6 |
| Rovinari | Rovinari city | 5–6 |
| Pinoasa | | 2–3 |
| Roșia de Jiu | | <1 |
| North Peșteana | | 10–12 |
| | Peșteana Jiu, Valea cu Apă, Hotăroasa, Bălteni, Cocoreni, Urdari villages | ≈1 |

### 3.1.8. Investments for Land Recovery and Rehabilitation (C8)

To evaluate the costs of land reclamation, the possibilities of flooding, the necessary measures for stabilization, modeling, revegetation of the land, and the possibilities of reuse, modernization of existing constructions (buildings, halls, warehouses, etc.) or the construction of new objectives were taken into account and it was found that in the case of the Tismana and Pinoasa open-pits medium investments are required, while in the case of the other open-pits small investments are required.

### 3.1.9. Population Requirements (C9)

The assessment of the flooding opportunity of the lignite open-pits from the Rovinari mining basin was made taking into account the partial results obtained following an online survey, at the regional level (Figure 8). The questionnaire includes 5 questions regarding the possibility of reusing former quarries in the Rovinari mining basin. The needs of individuals and local communities, the importance of the socio-economic development of the region, and the recovery of the environment are taken into account, all these aspects being seen from the point of view of the respondent. Given the possibilities of reusing the former open-pits after the cessation of the mining activity depending on the characteristics of the site, five options were offered: landfill, pit lake, agriculture, forestry, and industrial museum or other cultural attractions [37].

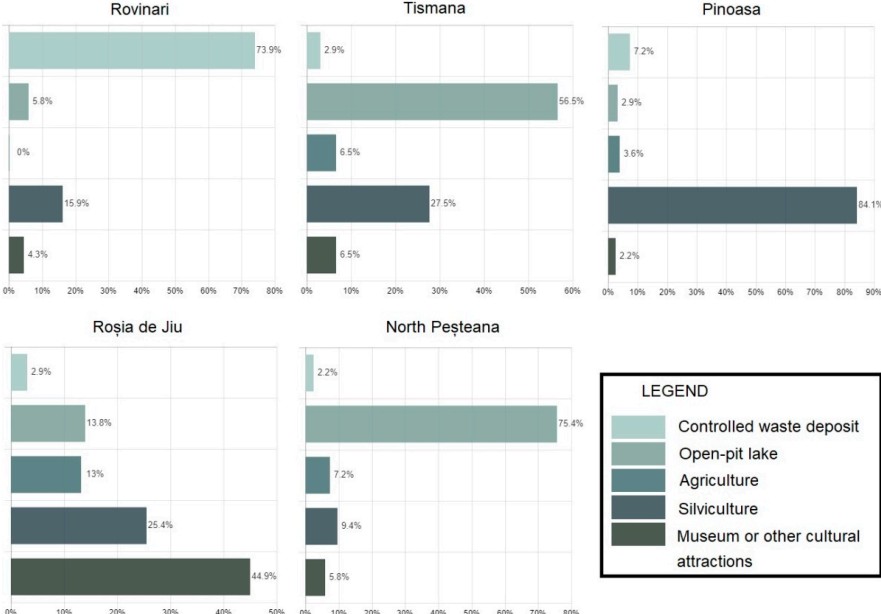

**Figure 8.** Partial survey results.

Depending on the type of reuse of major interest in assessing the opportunity of flooding, namely "open-pit lake" and its position in the hierarchy of population requirements, the corresponding score was calculated (using Equations (2) and (3)).

### 3.2. Final Evaluation of Flooding Opportunity

The scores obtained for the Rovinari lignite open-pits according to the assessing criteria allowed the construction of the final evaluation matrix (Table 5).

**Table 5.** Final matrix for assessing the opportunity of flooding of Rovinari lignite open-pits.

| Perimeter Criteria | Tismana | Rovinari | Pinoasa | Roșia de Jiu | North Peșteana |
|---|---|---|---|---|---|
| C1—Geomorphology and orography of the area | 2 | 2 | 1 | 1 | 3 |
| C2—Configuration of the remaining gap | 3 | 3 | 1 | 1 | 3 |
| C3—Necessity to restore the aquifer resources | 2 | 3 | 3 | 2 | 3 |
| C4—Necessity of appearance of a water body in the area | 3 | 3 | 3 | 3 | 3 |
| C5—Hydrology and hydrogeology of the region | 0 | 2 | 1 | 3 | 3 |
| C6—Stability conditions of the final slopes | 1 | 1 | 3 | 3 | 3 |
| C7—Accessibility and distance to the areas of interest | 2 | 2 | 2 | 3 | 3 |
| C8—Investments for land recovery and rehabilitation | 1 | 2 | 1 | 2 | 2 |
| C9—Population requirements | 3 | 1.5 | 0.75 | 1.5 | 3 |
| FINAL SCORE [1] (average mean) | 1.89 | 2.25 | 1.875 | 2.25 | 2.89 |

[1] 0 ÷ 0.99—inopportune; 1 ÷ 1.99—reduced opportunity; 2 ÷ 2.49—average opportunity; 2.5 ÷ 3—high opportunity.

The final scores were determined by calculating the arithmetic mean, and based on this score, a hierarchy of opportunity of flooding of the remaining gaps/former open-pits was established.

As a result of the assessment, based on the established assessment scale, the lignite open-pits from the Rovinari mining basin present the following flooding opportunity: North Pesteana open-pit-high/major opportunity, Rovinari and Roșia de Jiu open-pits-average opportunity and Tismana and Pinoasa open-pits-reduced opportunity of flooding.

## 4. Conclusions and Final Remarks

The methodology developed for assessing the flooding opportunity of former open-pits has a general character, can be applied to any type of open-pit in the world, and is an extremely useful tool for choosing the type of reuse of degraded mining lands, which can be used by both the economic operator and the companies designing the rehabilitation works.

The usefulness of this model consists mainly in the fact that it considerably reduces the risk of improper recovery and reuse of former open-pits in relation to the existing local conditions and the real requirements of the region and local communities. Although in many cases the creation of a lake in the post-mining landscapes is the desired solution, the concrete conditions are not always favorable. For example, even if the flooding of the remaining gap is possible in terms of available water resources, there is a risk of occurrence of negative effects from a geotechnical or ecological point of view in the medium and/or long term. Based on various assessing criteria, the methodology ensures a proper evaluation of the existing conditions and an appropriate assessment guiding the open-pits that are

suitable for flooding, while ensuring a high degree of security and safety for the objectives in the areas of influence.

Moreover, the methodology can be transposed into a telephone application or software, being very useful in this age of technology, which ensures a quick analysis of information providing feedback on the opportunity of flooding the analyzed open-pit.

The main advantage of the proposed model consists in the fact that it ensures an efficient evaluation (due to a large number of criteria it takes in the analysis) of former open-pits to establish the direction of sustainable recovery and reuse of the mining degraded land. This model ensures a step-by-step evaluation, in a relatively simple way, like a guide, but in the form of a matrix.

The most probable aspects and situations that can be found in situ have been transposed into opportunities, in this case, flooding opportunities.

The final matrix can be used exclusively to analyze the opportunity of flooding of former open-pits. It offers the possibility to obtain indications regarding the flooding opportunity of former open-pits in a short time. However, a more detailed analysis can be done following the entire proposed model and if necessary, adapting some aspects to the specific situation.

Most of the negative consequences of an erroneous decision are eliminated as a result of the application of such a methodology, which takes into account a complex set of assessing criteria and a sufficient number of options. The more criteria are fulfilled, in the favor or to the detriment of the flooding and the creation of an open-pit lake, the more certain it is to obtain an optimal result, and implicitly the mistakes in determining the direction of recovery of a former open-pit are insignificant and can be neglected.

Any former open-pit in the world can be evaluated based on these criteria and the final matrix. Taking into account the many open-pits (former, still operating, and future ones) that result in remaining gaps that have a negative impact on the environment, we believe that such a model for evaluation of flooding opportunity can be widely used, simplifying the planning process of degraded mining lands. In some situations, it is possible that aspects such as water quality are essential, at least in establishing the direction of reuse (because a lake with acid waters cannot be used for irrigation or swimming, but it can be an excellent tourist attraction or a location for conducting scientific research; the situation does not prevent the flooding of the remaining gaps of former open-pits if the flooding conditions are favorable, but it limits the uses of the pit lake) and this improvement can also be made by students, professors, researchers, or engineers interested in this problem.

By applying the methodology of a case study, it proved to be effective and the results obtained were plausible and applicable. As seen in the final matrix, North Peșteana open-pit presents a high opportunity of flooding as a result of favorable existing conditions, such as: development in the Jiu river meadow area, high depth of the open-pit (of 80 m), reduced investments since there are stable slopes, important inflow of water from precipitation and aquifer formations which contributes to the natural flooding of the open-pit, constructions which can take other functions, short distances to the surrounding localities, necessity of restoration of aquifer resources and of occurrence of a water body in the area, which have special advantages in terms of development and maintenance of newly installed vegetation on the degraded land, crops, orchards, restoration of water resources for local communities, etc., and last but not least, compliance with the population requirement since the open-pit lake can take important functions such as lake for leisure and recreation or water reservoir for crops irrigation, especially in drier periods. Tismana and Pinoasa open-pits present a reduced opportunity for flooding primarily as a result of hydrological and hydrogeological conditions which are not favorable for flooding to which are added, depending from one case to another, instability of the slopes, reduced depth of the pit, population requirement, medium distances to the city, etc., requiring high investments for the creation of open-pit lakes, so it is recommended to choose another type of reuse for the two open-pits.

The occurrence of a water body, in an area that has never met such conditions before, implies microclimatic changes due to the increase of the amount of evaporated water. The vapors reach the

atmosphere where they contribute to the formation of clouds, which in turn generate higher amounts of precipitation in the region or the surrounding areas, depending on the atmospheric circulation. Microclimatic changes can be a disadvantage because they contribute to the accentuation of the greenhouse effect, global warming, intensification of extreme phenomena, more violent storms, but for areas with a rainfall deficit, these changes are an advantage.

The advantages of creating open-pit lakes vary depending on the final type of reuse of the lake, so several common advantages are presented: the rehabilitation and reintegration of the degraded land into the landscape; the restoration of the aquifer resources, especially of the phreatic layer with advantages for the development and maintenance of the newly installed vegetation, crops, etc.; establishment of a new ecosystem, an aquatic one, and specific biodiversity; the restoration of adjacent ecosystems and local biodiversity; the restoration of drinkable water resources; source of water for irrigation of adjacent agricultural lands; sustainable development of the area; economic development (tourism, pisciculture) and so on.

**Author Contributions:** Conceptualization, I.-M.A., M.L., and F.F.; methodology, I.-M.A.; software, I.-M.A., M.L., and F.F.; validation, M.L. and F.F.; formal analysis, I.-M.A., M.L., and F.F.; investigation, I.-M.A. and M.L.; resources, I.-M.A., M.L., and F.F; writing—original draft preparation, I.-M.A.; writing—review and editing, M.L. and F.F.; visualization, I.-M.A., M.L., and F.F.; supervision, M.L. and F.F.; project administration, I.-M.A.; funding acquisition, I.-M.A., M.L., and F.F. All authors have read and agreed to the published version of the manuscript.

**Funding:** The APC was funded by the RAFF project (Risk Assessment of Final Pits During Flooding) co-financed by the Research Fund for Coal and Steel (RFCS) under the Grant Agreement No-847299-RAFF.

**Acknowledgments:** The research presented in this paper has been conducted within the RAFF project (Risk Assessment of Final Pits During Flooding) co-financed by the Research Fund for Coal and Steel (RFCS) under the Grant Agreement No-847299-RAFF.

**Conflicts of Interest:** The authors declare no conflict of interest. The funders had no role in the design of the study; in the collection, analyses, or interpretation of data; in the writing of the manuscript, or in the decision to publish the results.

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
