# Peer review of "A Model to Evaluate the Flooding Opportunity and Sustainable Use of Former Open-Pits"

_sustainability, doi:10.3390/su12219275_

Round 1

Reviewer 1 Report

In order to improve the article, it would be necessary to carry out the following:

          -Improving the quality of the figures 1, 2 and 8.

Reviewer 2 Report

Interesting article dealing with the significant problem of flooding arising. Opencast lakes and their flooding, an interesting way of reclamation and the use of degraded areas. The study developed a methodology for assessing this possibility with the aim of looking for open trenches that are susceptible to flooding.
To this end, more criteria, detailed determination complexes, allowing for the option of flood occurrence. The methodology is also aimed at ensuring maximum safety conditions on post-mining sites, meeting local socio-economic and cultural conditions, harmonization with neighboring ecosystems and sustainable development of the region.
Reviewer's suggestions:
- reflecting on the title of the work - comparing that it refers to an answer or methodology
- presentation of fig. 5 - in a clearer way
- Fig. 8 - another form of deletion: other - at the moment the data is not legible
- tab. 2 should not be divided
- the title has been written generally (you can suggest a more specific title, e.g. "Model ...")

Reviewer 3 Report

please refer to the attached document!

Round 2

Reviewer 3 Report

good morning dear authors,
there was probably an error in sending my original report. Only part of it has come to you. I am sending you the original report

Round 3

Reviewer 3 Report

Please, refer to the attached document!
